# Antioxidant Activity of Polysaccharides from the Edible Mushroom *Pleurotus eryngii*

**DOI:** 10.3390/molecules28052176

**Published:** 2023-02-26

**Authors:** Tania Petraglia, Tiziana Latronico, Angela Fanigliulo, Aniello Crescenzi, Grazia Maria Liuzzi, Rocco Rossano

**Affiliations:** 1Department of Sciences, University of Basilicata, 85100 Potenza, Italy; 2Department of Biosciences, Biotechnologies and Environment, University of Bari “Aldo Moro”, 70126 Bari, Italy; 3Bioagritest Srl-Centro Interregionale di Diagnosi Vegetale, 85010 Pignola, Italy; 4School of Agricultural, Forestry, Food and Environmental Sciences, University of Basilicata, 85100 Potenza, Italy

**Keywords:** mushrooms, β-glucan, *Pleurotus eryngii*, polysaccharides, antioxidant activity, neuroprotection

## Abstract

In this study the antioxidant and neuroprotective activity of an enriched polysaccharide fraction (EPF) obtained from the fruiting body of cultivated *P. eryngii* was evaluated. Proximate composition (moisture, proteins, fat, carbohydrates and ash) was determined using the AOAC procedures. The EPF was extracted by using, in sequence, hot water and alkaline extractions followed by deproteinization and precipitation with cold ethanol. Total α- and β-glucans were quantified using the Megazyme International Kit. The results showed that this procedure allows a high yield of polysaccharides with a higher content of (1-3; 1-6)-β-D-glucans. The antioxidant activity of EPF was detected from the total reducing power, DPPH, superoxide, hydroxyl and nitric oxide radical scavenging activities. The EPF was found to scavenge DPPH, superoxide, hydroxyl and nitric oxide radicals with a IC50 values of 0.52 ± 0.02, 1.15 ± 0.09, 0.89 ± 0.04 and 2.83 ± 0.16 mg/mL, respectively. As assessed by the MTT assay, the EPF was biocompatible for DI-TNC1 cells in the range of 0.006–1 mg/mL and, at concentrations ranging from 0.05 to 0.2 mg/mL, significantly counteracted H_2_O_2_-induced reactive oxygen species production. This study demonstrated that polysaccharides extracted from *P. eryngii* might be used as functional food to potentiate the antioxidant defenses and to reduce oxidative stress.

## 1. Introduction

Mushrooms are edible fungi that possess high nutritional value and many biological activities such as immunomodulatory, hepatoprotective, antitumor, anti-inflammatory, antiviral, hypoglycemic, hypolipidemic and antioxidative [1,2,3,4,5,6]. Mushrooms belong to the phylum Basidiomycota, which includes more than 2000 edible species [7]. *Pleurotus* species represent one of the mushroom species with the highest production in the world. Among the *Pleurotus* genus, *Pleurotus eryngii* (known as king trumpet or king oyster mushroom), besides its excellent taste and flavor, represents one of the most commercially important species in the world due to the capability to efficiently degrade many agricultural wastes and grow at a wide range of temperatures [8]. Additionally, it possesses many biological activities such as antitumor, antioxidative, immunomodulatory, hepatoprotective, anti-inflammatory, antiviral, hypoglycemic, hypolipidemic, hypocholesterolemic and fibrinolytic [3,9,10,11,12,13,14,15]. Due to the significant content of proteins, glucans, fibers, unsaturated fatty acids, phenolic compounds, vitamins, minerals and secondary metabolites associated with a low lipid content, *P. eryngii* have been appreciated for their nutritional value [8,16]. Cultivated oyster mushrooms of the genus *Pleurotus* are interesting as a source of biologically active glucans with a variety of bioactivities, with potential use in the field of biomedical sciences [17]. In particular, many of the biological activities of *P. eryngii*, such as immunostimulating, antitumor, hepatoprotective, antioxidant, hypolipidemic and hypoglycemic, are attributable to polysaccharides [9,13,14,18,19,20,21,22]. β-glucans from *Pleurotus* sp. (pleuran) have been used as food supplements [23]. Mushroom polysaccharides are present mostly as linear and branched glucans with different types of glycosidic linkages, such as (1,3-1,6)-β-glucans and (1-3)-α-glucans [24], but some are true heteroglycans containing glucuronic acid, xylose, galactose, mannose, arabinose or ribose [25,26]. The physicochemical properties of β-glucans differ depending on the characteristics of their primary structure, including the linkage type, the degree of branching, the molecular weight, and the conformation [27]. Many mushroom polysaccharides, isolated from the fruiting body [22,28,29,30] or submerged culture of *P. eryngii* [31], have been reported to have significant antioxidant activities based on various in vitro and in vivo assays [28,31,32]. The antioxidant activity of polysaccharides is attributed to the ability of radical scavenging, to chelate Fe^2+^, to inhibit lipid peroxidation and, finally, to increase the activity of antioxidant enzymes such as superoxide dismutase, catalase and glutathione peroxidase [5,33]. To this purpose, Kishk and Al-Sayed [34] reported that the radical scavenging mechanism of polysaccharides is similar to that of phenol compounds through hydrogen atom transfer, in the case of neutral polysaccharides, and through the electron transfer mechanism, usually occurring for the acidic polysaccharides. The antioxidant activity of polysaccharides is of particular interest as numerous studies have highlighted their ability in reducing oxidative stress induced by an excessive production of reactive oxygen species (ROS). Thanks to this property, polysaccharides could mitigate the toxicity associated with ROS overexpression and provide a potential therapeutic approach in the prevention of various diseases [35,36]. Furthermore, the interest in the biological activity of polysaccharides has increased since both in vitro and in vivo studies demonstrated that they possess neuroprotective activities, justifying their use for the prevention of neurological diseases and for the improvement of the effectiveness of conventional treatments [37,38,39]. Here, an enriched polysaccharide fraction (EPF) from the powdered fruiting body of *P. eryngii* was extracted and used in different assays to characterize its antioxidant and neuroprotective properties. The results of this study suggest that this mushroom, besides its nutritional value [16], possesses radical scavenging activity and, therefore, it might be used directly as functional food through dietary supplementation to reduce the level of oxidative stress.

## 2. Results and Discussion

### 2.1. Proximate Composition

Table 1 reports the proximate composition of the fruiting body of cultivated *P. eryngii*, expressed as both a percentage of fresh weight and dry mass. The data confirm that *P. eryngii* is a low-calorie food since it brings low amounts of fats. These findings are in agreement with those described by Roncero-Ramos et al. [40] for this species of mushroom harvested from cultivation rooms. By contrast, as reported by Manzi et al. [16] for raw commercial samples of *P. eryngii*, the proximate composition differs for the nutritional value since they found a lower moisture value but higher lipid, ash, carbohydrate and protein content than our study. However, it is known that the growth material can influence the chemical composition and, as a consequence, the nutritional value of the cultivated mushroom.

### 2.2. Glucan Content

The total α- and β-glucans were determined using the assay kit of Megazyme International. As shown in Table 2, the fruiting bodies contained 28.37 ± 0.92 g/100 g dry mass of total glucans, of which 2.15 ± 0.18 g/100 g dry mass (7.58 ± 0.63%) and 26.22 ± 0.73 g/100 g dry mass (92.42 ± 2.57%) corresponded to α- and β-glucans, respectively. The total glucan content found in the present study was lower than that reported by Roncero-Ramos et al. [40], but higher than the values reported by Sari et al. [41] for commercially cultivated *P. eryngii*. In both studies, glucans were measured using the same assay used in this work. According to various studies, the majority of glucans present in *P. eryngii* are beta-type; indeed, α-glucans (phytoglycogen) and starch are usually low in commonly cultivated mushrooms, less than approximately 10% [23]. However, the glucan content may vary according to the topological specificity in the fruit bodies, the individual strains and the assay used for its detection. Different methods have been described to measure the content of glucans extracted from mushrooms. Mc Cleary and Draga [42] compared different procedures and developed a reliable, quantitative method for the measurement of β-glucans in mushroom and mycelial products. This method is based on controlled acid hydrolysis of total glucans, the measurement of α-glucans using an enzymatic procedure and the detection of β-glucans by difference. In comparison with other enzymatic procedures for the direct measurement of β-glucans, the method described by Mc Cleary and Draga [42], which resembles the method used in the present work, allowed the detection of amounts of total and β-glucans in their samples from *P. eryngii* higher than those detected in our samples, whereas the amount of α-glucans was smaller. By contrast, Manzi et al. [16,25] described another procedure based on the direct hydrolysis by lichenase and β-glucanase which allowed detecting the content of β-glucan in the fruit bodies of *P. eryngii* about 1–2 orders lower than that measured in our samples. These authors attributed the low glucan content to the presence of inert, insoluble material in the fiber residue, which prevents the diffusion of enzymes during the β-glucan measurement.

### 2.3. Extraction of Polysaccharides from the Edible Mushroom P. eryngii

An enriched polysaccharide fraction (EPF) was obtained from the powdered fruiting body of *P. eryngii* by using, in sequence, hot water and alkaline extractions, followed by deproteinization and precipitation steps (Figure 1).

The extraction yield was 26.26 ± 0.93 g/100 g of dry mass. Chemical characterization (Table 3) showed that EPF is constituted of about 90% glucans and smaller amounts of proteins (3.55 ± 0.41%) and uronic (glucuronic and galacturonic) acids (2.24 ± 0.23%), whereas polyphenols were not found. These findings suggest that the procedure used allows a high recovery of glucans (83% of the initial glucans were measured in EPF) and, at the same time, a low content of interfering substances. In this regard, a protein assay, carried out with the Bradford method [43], revealed the presence of small amounts of proteins, indicating that the application of two deproteinization steps through the Sevag reagent allowed effective protein removal, thus suggesting that the residual proteins detected could be due to their existence as a polysaccharide–protein complex [44]. To exclude the residuals of major antioxidants such as phenolic compounds from polysaccharide extract, the Folin–Ciocalteu test was conducted. The results shown in Table 3 indicate the absence of phenolic compounds, suggesting the efficiency of the ethanol treatment as the initial step of the procedure and dialysis. The estimation of the molecular weight range of polysaccharides in EPF, determined using an ultrafiltration column, revealed that the main glucan fraction (about 80%) was in the range of molecular weight 10–50 kDa.

About 20% of the glucans were estimated to be greater than 50 kDa, whereas no glucans were detected below 10 kDa (Table 4). To this purpose, recently, Ellefsen et al. [45] reported on three water-soluble *P. eryngii* polysaccharide fractions, one neutral of 16.7 kDa and two acidic of 22.8 and 17.2 kDa, respectively.

As shown in different studies, the yield of polysaccharides extracted from *P. eryngii* may vary in response to the method used for their extraction. As reported by Barbosa et al. [46], in addition to the hot aqueous extraction, which represents the most widely used technique for polysaccharide extraction [47], there are other polysaccharide extraction techniques such as microwave-assisted extraction, pulse extraction, ultrasound-assisted extraction, alkaline extraction, enzyme-assisted extraction, subcritical water extraction and the use of deep eutectic solvents, each of them having advantages and disadvantages. The extraction yield obtained in this study was slightly lower than that reported by Xu et al. [48], whereas it was much higher when compared with the 7.31% and 12.18% obtained by Li and Shah 2016 [49] and by He et al. [50], respectively. In the first two cases, the polysaccharide extraction was performed with hot water (100 °C × 120 min and 95 °C × 180 min), whereas in the latter, it was with hot alkaline extraction (90 °C × 300 min). Based on these data, it would appear that the procedure used in this study, based on hot water and alkaline extractions in sequence, allows obtaining a higher yield of polysaccharides. However, it is worth mentioning that Zhang et al. [22], using a response surface methodology to study the effects of ultrasonic power, ultrasonic time and the ratio of material to liquid on the extraction yields of *P. eryngii* polysaccharides, reported a yield of polysaccharides reaching 34%.

### 2.4. Antioxidant Activity

The antioxidant potential of polysaccharide extracted from the *P. eryngii* fruiting body was evaluated using various antioxidant assays, including total reducing power, 2,2-diphenyl-1-picrylhydrazyl (DPPH), superoxide, hydroxyl and nitric oxide radical scavenging activities. As shown in Figure 2, EPF exhibited antioxidant activity in a concentration-dependent manner. These results indicated the ability of EPF to scavenge DPPH, superoxide, hydroxyl and nitric oxide radicals with IC_50_ values of 0.52 ± 0.02, 1.15 ± 0.09, 0.89 ± 0.04 and 2.83 ± 0.16 mg/mL, respectively.

In addition, the reducing power at an EPF concentration of 10 mg/mL, expressed as absorbance at 700 nm, reached a value of 0.8 (Figure 3) with a rapid increase at concentrations above 4 mg/mL. ROS, as the superoxide anion and hydroxyl radical, are generated physiologically in the course of mitochondrial oxidative metabolism as well as in the cellular response to either xenobiotics, cytokines or microbial invasions. ROS and RNS (reactive nitrogen species) levels increase when their production overcomes the cellular antioxidant defenses, leading to oxidative stress which contributes to the development of several diseases and aging [51]. Several studies indicated that the polysaccharides from Basidiomycetes, and in particular from the *Plerotus* species, displayed strong in vitro free radical scavenging activity [28,52,53]. The comparison of results reported in this study with those obtained by other authors suggests that the percentage of inhibition of DPPH, superoxide and hydroxyl radical scavenging activities is not strictly dependent on the procedure used for the extraction of polysaccharides. Indeed, using concentrations of extract in the same range, but different methods of extraction of polysaccharides, both Zhang et al. [22] and Zhao et al. [30] found results similar to those reported in this study. 

In the present study, EPF showed scavenging activities towards four different free radicals. For all radicals, a relationship between the concentration of EPF and the antioxidant activity was observed. The free radical scavenging capacity of EPF may be related to the presence of hydrogen and hydroxyl groups in the polysaccharide chain, acting with a mechanism similar to that of phenol compounds [34]. However, even if the EPF showed a low content of interfering substances and the absence of major antioxidants such as phenolic compounds, the contribution of residual substances such as proteins and peptides with certain free radical scavenging activities cannot be excluded. The ubiquitous nitric oxide (NO·), a reactive nitrogen species (RNS), can be produced by three isoforms of NO synthase in the brain. It is an important highly diffusible biological messenger that plays a prominent role in the physiology of the CNS [54]. However, it is widely accepted that its overproduction leads to oxidative stress which causes significant damage to cell structures and biomolecular function, directly or indirectly leading to a number of neurological diseases [55]. As reported by several works, polysaccharides from mushrooms are considered effective free radical scavengers; however, in the literature there are no reports in which polysaccharides from mushrooms have been tested against RNS. Therefore, the finding of the antioxidant activity of EPF towards NO represents a novelty of this study.

### 2.5. Effect of EPF on DI-TNC1 Viability and ROS Production

Mushroom polysaccharides are increasingly attracting the scientific community for their possible use in the therapeutic field; indeed, they possess potential pharmacological activities with multi-targeting bioactivity, synergistic effects, low toxicity and high safety. Many studies have demonstrated that natural bioactive polysaccharides have neuroprotective activity against oxidative stress, apoptosis, neuroinflammation [56]. In the present study, the biocompatibility of the EPF and its neuroprotective potential were evaluated on DI-TNC1. This cell line was established from primary cultures of type 1 astrocytes from rat brain tissue [57]. The choice of these cells derived from the important role exerted by astrocytes for the maintenance of brain homeostasis. As reported in Figure 4a, there was no significant difference between the control group and samples with EPF at all the concentrations used (*p* > 0.05), suggesting that polysaccharides did not exhibit any cytotoxic effect up to 1 mg/mL of EPF. We also evaluated the effect of EPF towards oxidative stress and found that the EPF had a protective role against ROS production in DI-TNC1. Indeed, as reported in Figure 4b, at concentrations ranging from 0.05 to 0.2 mg/mL, the EPF was able to counteract ROS production induced by the harmful insult of H_2_O_2_. ROS have an important influence on the induction and development of neuroinflammation [58]. The central nervous system (CNS) is particularly vulnerable to oxidative stress due to its high oxygen consumption, weakly antioxidant systems and the high content of biomacromolecules susceptible to oxidation. The generation of ROS and the subsequent oxidative damage are believed to be involved in the pathogenesis of neurological disorders such as Alzheimer’s disease (AD), Parkinson’s disease (PD), amyotrophic lateral sclerosis (ALS) and multiple sclerosis (MS) [59,60]. Therefore, the modulation of the events leading to excess ROS production from CNS resident cells might contribute to mitigating neuronal damage occurring during neurological diseases. Apart from the endogenous antioxidant defense mechanisms, a very important source of antioxidants is represented by diet. As reported in numerous papers [61,62,63,64], antioxidant supplements or foods containing antioxidant compounds can be used to help the organism to reduce oxidative damage and, hence, to prevent those pathologies in which oxidative stress can play a fundamental role.

## 3. Materials and Methods

### 3.1. Chemicals

All reagents used were of the highest grade and were purchased from Sigma-Aldrich (St. Louis, MO, USA), Carlo Erba (Milan, Italy), Bio-Rad Laboratories (Segrate, Italy) and GE Healthcare (Uppsala, Sweden). The DI-TNC1 (ATCC CRL-2005) cell line, established from primary cultures of type 1 astrocytes from brain diencephalon tissue of 1-day-old rats [39], was acquired and authenticated from the ATCC (www.lgcstandards-atcc.org). Dulbecco’s modified Eagle’s medium (DMEM), trypsin, penicillin and streptomycin were provided by GIBCO (Paisley, Scotland). 2′,7′-dichlorofluorescein diacetate (DCFH-DA) was purchased from Calbiochem (San Diego, CA, USA). 3-(4,5-dimethylthiazol-2-yl)-2.5-diphenyltetrazolium bromide (MTT) was provided from Sigma (St. Louis, MO, USA).

### 3.2. Samples of P. eryngii Fruiting Bodies

*P. eryngii* fruiting bodies were produced in polypropylene bottles (1 L). The cultivation material consisted of 16.25% wheat straw, 15% beet, 3.75% CaCO_3_ and 65% water. The bottles filled with this material were sealed with seal caps made ventilated by inserting a polyester filter in the central part of the cap and were autoclaved at 121 °C and 16 psi for 30 min. Once sterilized, the bottles were allowed to cool in an aseptic environment in order to stabilize the temperature of the substrate. They were subsequently inoculated in a sterile environment with solid mycelium of the *P. eryngii* isolate AF72, grown on acidified potato dextrose agar medium (PDA) amended with streptomycin sulfate (0.02%), and incubated in a fridge thermostat for 25 days at 23 °C with 50 lux lighting for 6 h a day. After the surface of substrate was totally colonized with mycelium, in order to stimulate its differentiation, bottles were incubated at 10 °C for 10 days. This incubation favored the formation of the pinheads. Once the caps were removed, polypropylene bottles were incubated in a conditioned chamber with the following controlled environmental parameters: a relative humidity of 75–90%, a constant temperature of 22 °C (night/day), a constant brightness of 150 lux, and a CO_2_ concentration of 300–900 ppm. After 8 days, the fruiting bodies were harvested.

### 3.3. Proximate Composition

Two different preparations of samples were analyzed in duplicate for chemical composition (moisture, proteins, fat, carbohydrates and ash) using the AOAC procedures [65]. In particular, protein content (N × 4.38) was estimated with the macro-Kjeldahl method; the fat was determined by extracting a known weight of powdered sample with petroleum ether using a Soxhlet apparatus; and the ash content was determined by incineration at 600 ± 15 °C, whereas total carbohydrates were calculated by difference. Finally, energy was calculated according to the following equation: Energy (kcal) = 4 × (g protein + g carbohydrate) + 9 × (g fat).

### 3.4. Glucan Measurement 

The measurement of glucans in powdered fruiting bodies was carried out using the assay Kit “Mushroom and yeast β-glucan” K-YBLG of Megazyme International (Bray, Wicklow County, Ireland), according to the manufacturer’s instructions. Briefly, the total glucans were measured using controlled acid hydrolysis with H_2_SO_4_ followed by incubation with a mixture of exo-1,3-β-glucanase and β-glucosidase and the glucose released specifically was measured using glucose oxidase/peroxidase reagent. α-glucans were specifically measured after the hydrolysis of starch-like glucans to glucose with glucoamylase and the glucose specifically measured with a glucose oxidase/peroxidase reagent. β-glucans were determined by the difference between glucose contents after acidic hydrolysis of total glucans and specific enzymatic hydrolysis of α-glucans. In both determinations of the total glucan/α-glucan contents as well as the D-glucose in oligosaccharides, sucrose and free D-glucose contents were measured. The glucan content was expressed as a percentage of dry mass [42].

### 3.5. Preparation of Crude Polysaccharide Fraction

The procedure for the preparation of crude polysaccharide fraction (EPF) is shown in Figure 1. The residue obtained after the treatment of the powdered fruiting body with 80% ethanol (1:10, *w*:*v*) was extracted two times, first with distilled water at 100 °C for 6 h (water-soluble polysaccharide fraction) and then with 0.5 N NaOH at 4 °C for 6 h. After centrifugation (6000× *g*, 15 min, 4 °C), the alkali soluble polysaccharide fraction (ASPF) was first neutralized with 2N HCl and then combined with the water-soluble polysaccharide fraction (WSPF). After polysaccharide extraction, the sample was subjected to two deproteinization steps with Sevag reagent (butanol–chloroform, 1:4, *v*:*v*) with a ratio of polysaccharide fraction to Sevag reagent of 3:1 [66]. The polysaccharides were precipitated overnight at 4 °C adding four volumes of 95% ethanol. Following centrifugation (6000× *g*, 15 min, 4 °C), the precipitate was washed two times with cold acetone, re-dissolved in water and dialyzed (Cut-off 3 kDa) against water, then lyophilized. The total glucan content in EPF was measured as described before (see par. 3.4). The uronic acid content was determined according to the method of Blumenkrantz and Asboe-Hansen [67] using D-galacturonic acid as standard whereas proteins and polyphenols were measured according to the method of Bradford [43] and Folin–Ciocalteu [68].

### 3.6. Determination of Molecular Weight Range of Glucans

The molecular weight ranges of glucans in EPF were determined using the Vivaspin 6 ultrafiltration column (GE Healthcare) with a molecular weight cutoff (MWCO) of 50 and 10 kDa, respectively. An amount of 3 mL of the water-resuspended polysaccharide sample was applied on the 50 kDa ultrafiltration column, obtaining one retained fraction containing polysaccharides with a molecular mass greater than 50 kDa, while the pass-through was subsequently applied on the 10 kDa filters from which two other fractions were obtained, one containing polysaccharides with a mass between 10 and 50 kDa and the other containing polysaccharides smaller than 10 kDa. The total glucans in each fraction were determined as described above.

### 3.7. DPPH Radical Scavenging Activity

The capacity of polysaccharides to scavenge 2,2-diphenyl-1-picrylhydraziyl radical (DPPH˙) was evaluated adding 0.5 mL of samples at different concentrations to 0.5 mL of 0.2 mM DPPH˙ in methanol. After 30 min of incubation at room temperature in the dark, the absorbance was measured at 517 nm. The results were expressed as percentage of DPPH radical scavenging activity in comparison to positive control (gallic acid) using the following equation:(%) = [(Abs_control_ − Abs_sample_)/Abs_control_] × 100.

Antioxidant activity was indicated as IC_50_, representing the sample concentration (mg/mL) required to scavenge 50% of DPPH free radicals.

### 3.8. Superoxide Radical Scavenging Activity

The superoxide radical scavenging activity was determined as reported by Larocca et al. [69]. An aliquot of 100 μL of polysaccharide fraction was added to 100 μL of 2 mM NADH and 100 μL of 50 µM nitrotetrazolium blue. Then, 100 μL of 50 µM phenazine methosulphate was added to the mixture, and after 5 min of incubation at room temperature the absorbance, was read at 560 nm. The results were expressed as scavenging activities in comparison to positive control (ascorbic acid) using the following equation:(%) = [(Abs_control_ − Abs_sample_)/Abs_control_] × 100.

Antioxidant activity was expressed as IC_50_, indicating the sample concentration (mg/mL) required to scavenge 50% of superoxide radicals.

### 3.9. Hydroxyl Radical Scavenging Activity

The hydroxyl radical scavenging activity was measured as described by Winterbourn and Satton [70] with some modifications. Briefly, 0.75 mL of EPF (at various concentrations) in sodium phosphate buffer (pH 7.4) was mixed with 0.5 mL of 0.3 mM safranin solution and 1 mL of 6 mM EDTA-Fe(II) solution. The reaction was initiated by adding 0.5 mL of 3% H_2_O_2_. After incubation at 37 °C for 30 min, the absorbance was read at 536 nm. The results were expressed as percentage of hydroxyl radical scavenging activity in comparison to positive control (gallic acid) using the following equation:(%) = [(Abs_control_ − Abs_sample_)/Abs_control_] × 100. 

Antioxidant activity was indicated as IC_50_, representing the sample concentration (mg/mL) required to scavenge 50% of DPPH free radicals.

### 3.10. Nitric Oxide Scavenging Activity

The nitric oxide radical scavenging activity was determined as described by Larocca et al. [62]. 100 μL of the sample solution was mixed with 100 μL of 20 mM sodium nitroprusside and incubated for 60 min at room temperature. Then, 100 μL of Griess reagent were added to the mixture and the absorbance was measured at 560 nm after 10 min of incubation in the dark. The results were expressed as scavenging activities in comparison to positive control (ascorbic acid) using the following equation:(%) = [(Abs_control_ − Abs_sample_)/Abs_control_] × 100 

Antioxidant activity was expressed as IC_50_, indicating the sample concentration (mg/mL) required to scavenge 50% of superoxide radicals.

### 3.11. Total Reducing Power

For the total reducing power assay, 0.1 mL of EPF at different concentration was mixed with 0.5 mL of 0.2 M sodium phosphate buffer at pH 6.6 and 0.5 mL of 1% potassium ferricyanide. After 30 min of incubation at 50 °C, 0.5 mL of 10% trichloroacetic acid was added to the mixture and then centrifuged (10 min at 4000× *g*). Finally, 0.5 mL of the supernatant was mixed with 0.5 mL of distilled water and 0.1 mL of 0.1% ferric chloride and the absorbance was read at 700 nm [71].

### 3.12. MTT Viability Assay

The effect of EPF on cell viability of DI-TNC1 cells was detected using the MTT [3-(4,5-dimethylthiazol-2-yl)-2,5-diphenyl tetrazolium bromide] assay as reported by Latronico et al. [72]. Briefly, confluent cells in serum-free DMEM were treated for 20 h with EPF at concentrations ranging from 0.006 to 1 mg/mL; then, the culture medium was removed and cells were incubated for 2 h at 37 °C, 5% CO_2_ with 0.5 mg/mL of MTT. At the end of incubation, the culture medium was removed and the formazan crystals in cells were solubilized with 90% ethanol. The amount of formazan product was determined by optical absorbance at 545 nm with a reference wavelength of 690 nm. Cell viability was expressed as percentage of control (CTRL), represented by untreated cells, which was set at 100%.

### 3.13. Intracellular Reactive Oxygen Species Detection

The detection of reactive oxygen species (ROS) in DI-TNC1 cells was performed as reported by Latronico et al. [73]. Briefly, confluent DI-TNC1 cells, seeded in 96-well plates, were pre-treated for 1 h with EPF at concentration ranging from 0.006 to 0.2 mg/mL, then loaded with 10 μM of the fluorescent probe 2′,7′-dichlorofluorescein diacetate (DCFH-DA) in phenol red–free DMEM. After 30 min of incubation at 37 °C, DCFH-DA was removed from wells and the cells were treated for 1 h with EPF at the same doses used in the pre-treatment, in the presence of 200 μM H_2_O_2_. The negative (CTRL) and positive controls (H_2_O_2_) were represented by cells treated only with DCFH-DA or with H_2_O_2_, respectively. To measure the fluorescence intensity, cells were subjected to spectrofluorimetric analysis at 525 nm under excitation at 485 nm in a microplate reader (Cytation 3 Imaging Reader; Bio Tek, Winooski, VT, USA). Results were normalized to cell viability and ROS production was expressed as the relative percentage of photoluminescence intensity (PLI) versus positive control using the following equation: % ROS production = (PLI_sample_/PLI_H2O2_) × 100.

## 4. Conclusions

In the present study, an enriched polysaccharide fraction (EPF) was obtained from *P. eryngii* fruiting bodies by using a procedure of extraction with hot water and alkaline solutions. Polysaccharide analysis revealed that this fraction was mainly composed of β-glucans and possessed antioxidant capacities of scavenging free radicals. The in vitro biocompatibility of EPF and the ability to counteract ROS, tested on an astrocyte cell line, suggested that this fraction may exert neuroprotective activity.

All these results suggest that polysaccharides from *P. eryngii* might be used directly as functional food to potentiate the antioxidant defenses and to prevent or attenuate diseased states, such as neurodegenerative diseases, in which the oxidative stress plays a fundamental role. However, more studies are required to purify this fraction and to explore its complete structural characteristics and the mechanism of its neuroprotective activity. 

The possibility of modulating in a laboratory the composition of the growth substrate can allow obtaining and selecting *P. eryngii* strains enriched in the polysaccharide components and in other molecules which may enhance the nutraceutical properties of this edible mushroom.

## Figures and Tables

**Figure 1 molecules-28-02176-f001:**
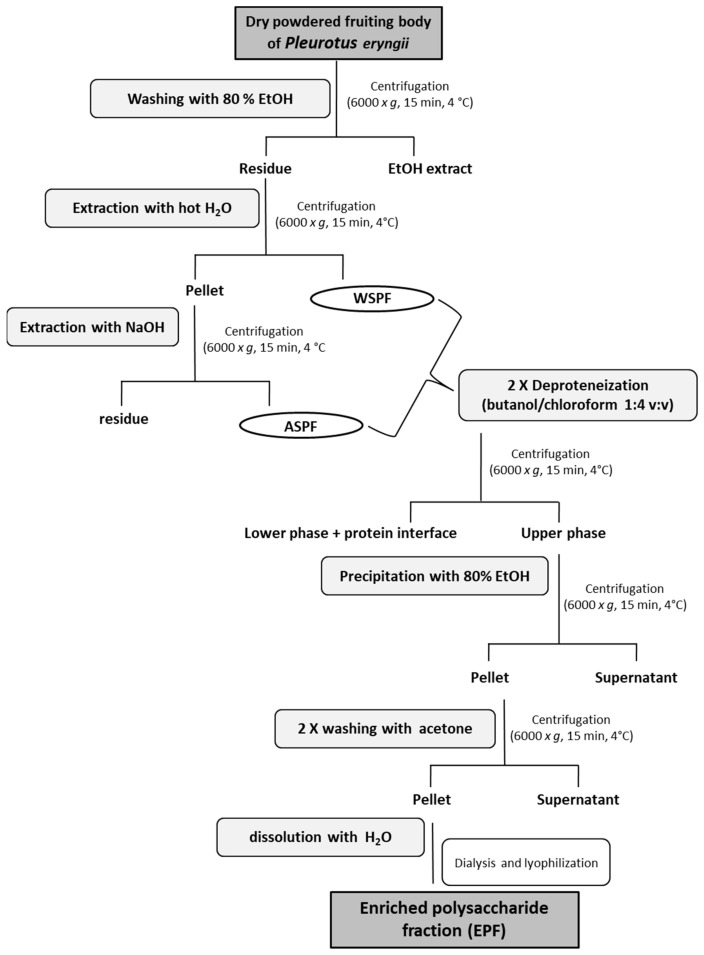
Schematic diagram depicting the preparation of the enriched polysaccharide fraction (EPF) from *P. eryngii*.

**Figure 2 molecules-28-02176-f002:**
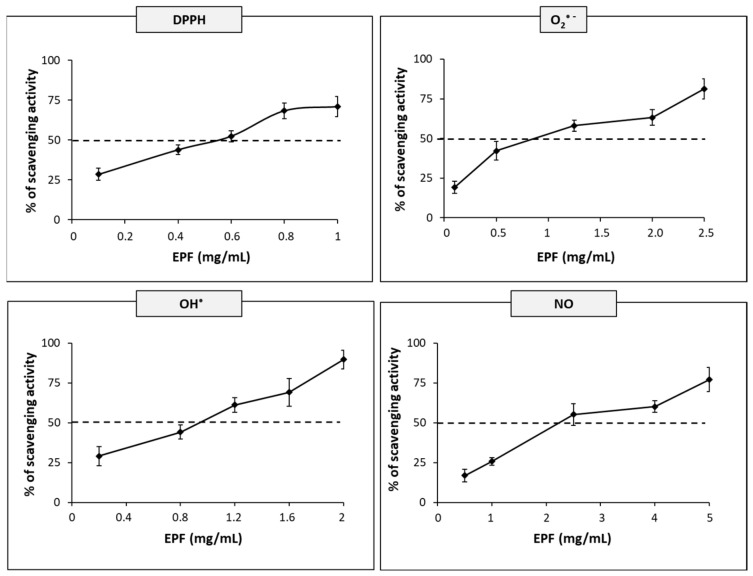
Scavenging activities of the enriched polysaccharide fraction from *P. eryngii*. The graphs represent the scavenging activities of the enriched polysaccharide fraction (EPF) against, 2,2-diphenyl-1-picrylhydrazyl (DPPH), superoxide radical (O_2_-^•^), hydroxyl radical (OH^•^) and nitric oxide (NO). Results are expressed as relative percentage of scavenging activity. Data are mean values ± SD of *n* = 3 experiments performed on different EPFs. Dashed lines indicate the IC_50_ values.

**Figure 3 molecules-28-02176-f003:**
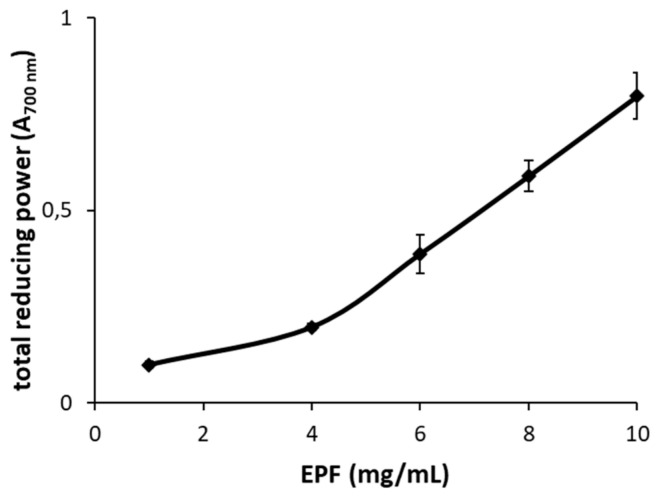
Total reducing power of the enriched polysaccharide fraction (EPF) from *P. eryngii.* Results are expressed as absorbance at 700 nm. Data are mean values ± SD of *n* = 3 experiments performed on different EPF.

**Figure 4 molecules-28-02176-f004:**
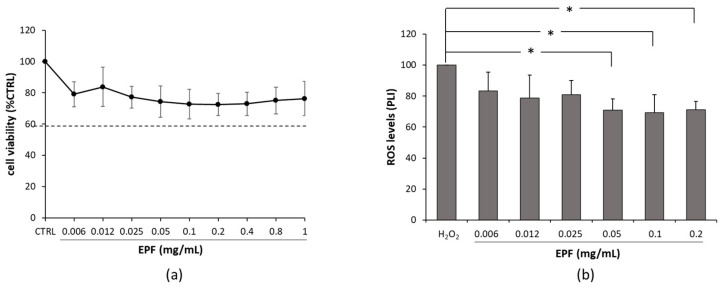
Effect of the enriched polysaccharide fraction on cell viability and reactive oxygen species (ROS) production in DI-TNC1 (ATCC CRL-2005) cell line. (**a**) Dose–response curve of cell viability, expressed as percentage of cell survival in comparison with control. The horizontal dashed line, set at 60%, indicates the threshold of cell viability. (**b**) ROS production is expressed as relative percentage of photoluminescence intensity (PLI) versus positive control (H_2_O_2_). Values are mean ± SD of *n* = 3 experiments performed on different cell populations. Statistically significant decrease in comparison with H_2_O_2_. is indicated by asterisks (one-way ANOVA followed by Dunnet’s post hoc test; * *p* < 0.05).

**Table 1 molecules-28-02176-t001:** Proximate analysis of *P. eryngii* fruiting body.

	Moisture	Proteins	Fat	Ash	Carbohydrates	Energy
(g/100 g fw) (g/100 g dm)	89.08 ± 0.98	1.36 ± 0.02 12.45 ± 0.18	0.17 ± 0.01 1.55 ± 0.09	0.71 ± 0.05 6.50 ± 0.46	8.68 ± 0.21 79.49 ± 1.92	* 41.69 ± 1.34 ** 381.78 ± 12.27

Values are reported as mean ± SD of two independent experiments performed in duplicate (*n* = 4). Fw: fresh weight; dm: dry mass; * kCal/100 g fw; ** kCal/100 g dm.

**Table 2 molecules-28-02176-t002:** Glucan content in powdered fruiting bodies of *P. eryngii*.

Total Glucans	α-Glucans	β-Glucans
(g/100 g Dry Mass)	(g/100 g Dry Mass)	(g/100 g Dry Mass)
28.37 ± 0.92	2.15 ± 0.18	26.22 ± 0.73
(100%)	(7.58 ± 0.63%)	(92.42 ± 2.57%)

Glucans were determined by the assay kit (Megazyme Cat. No. K-YBGL). β-glucans were determined by difference. Values are reported as mean ± SD of two independent experiments performed in triplicate (*n* = 6).

**Table 3 molecules-28-02176-t003:** Chemical characterization of the enriched polysaccharide fraction (EFP).

^a^ Yield	^b^ Glucan (%)	^b^ Protein (%)	^b^ Polyphenol (%)	^b^ Uronic Acid (%)
26.26 ± 0.93	89.59 ± 2.55	3.55 ± 0.41	n.d.	2.24 ± 0.23

^a^ Expressed as g of EFP/100g of dry mass. ^b^ Expressed as % of EFP. Values are reported as mean ± SD of two independent experiments performed in triplicate (*n* = 6).

**Table 4 molecules-28-02176-t004:** Estimation of molecular weight range of glucans in EFP.

Glucans < 10 kDa	10 kDa < Glucans < 50 kDa	Glucans > 50 kDa
(n.d.)	(79.61 ± 5.08%)	(17.54 ± 1.46%)

The molecular weight range of glucans was determined using the Vivaspin 6 ultrafiltration column (GE Healthcare) MWCO of 50 and 10 kDa, respectively. Values are reported as mean ± SD of three independent experiments (*n* = 3).

## Data Availability

Not applicable.

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
