# Peer review of "Antioxidant Activity of Polysaccharides from the Edible Mushroom Pleurotus eryngii"

_molecules, 2023, doi:10.3390/molecules28052176_

Round 1

Reviewer 1 Report

The authors carried out a study entitled "Antioxidant activity of polysaccharides from the edible mushroom Pleurotus eryngii", and these are my suggestion:

-     Please point out the novelty of the present study and future perspectives as well.

- Think about better option for keywords, instead cultivation. You can use the name of the analysed mushroom.

- Avoid using Our and We throughout the text.

- In the Introduction part please add some very important facts about Pleurotus eryngii.

- The discussion is very narrow and the results are insufficiently commented on and compared with the literature.

- You can really upgrade your investigation using advanced mathematical models. Statistical methods used in this paper are the biggest weakness of this study, but you could have an elegant solution for better research presenting the obtained results.

- Please, include correlation analyses for these results.

- In the Methods part, include the reference for the reducing power assay.

- The whole text needs to be revised (technical issues – coma, point, etc.).

- English editing required.

After revision of the paper, I think this is good work for publishing in this Journal.

Author Response

Reply to the reviewer 1

Review Report Form

English language and style

( ) English very difficult to understand/incomprehensible
( ) Extensive editing of English language and style required
(x) Moderate English changes required
( ) English language and style are fine/minor spell check required
( ) I don't feel qualified to judge about the English language and style

Yes

Can be

improved

Must be improved

Not

applicable

Does the introduction provide sufficient background and include

all relevant references?

( )

( )

(x)

( )

Are all the cited references relevant to the research?

( )

(x)

( )

( )

Is the research design appropriate?

( )

(x)

( )

( )

Are the methods adequately described?

( )

(x)

( )

( )

Are the results clearly presented?

( )

(x)

( )

( )

Are the conclusions supported by the results?

( )

(x)

( )

( )

Comments and Suggestions for Authors

The authors carried out a study entitled "Antioxidant activity of polysaccharides from the edible mushroom Pleurotus eryngii", and these are my suggestion:

- Please point out the novelty of the present study and future perspectives as well.

Reply: A sentence pointing out the novelty and the future perspectives of the study have now been added, respectively, to the results (paragraph 2.4,  lines 239-248,) and the conclusions (lines 469-475).

- Think about better option for keywords, instead cultivation. You can use the name of the analysed mushroom.

Reply: According to the suggestion of the reviewer in the keywords the term cultivation” has been replaced with “Pleurotus eryngii”.

- Avoid using Our and We throughout the text.

Reply: Done

- In the Introduction part please add some very important facts about Pleurotus eryngii.

Reply: According to the suggestion of the referee we have now added in the introduction other information about Pleurotus eryngi (lines 38-44).

- The discussion is very narrow and the results are insufficiently commented on and compared with the literature.

Reply: We thank the reviewer for his/her suggestions and we have now added more comments in the “result and discussion” section, comparing our results with the literature (paragraph 2.2, lines: 102-125; paragraph 2.3, 171-188; paragraph 2.4, 216-224 and 231-248; paragraph 2.5, 251-255).

- You can really upgrade your investigation using advanced mathematical models. Statistical methods used in this paper are the biggest weakness of this study, but you could have an elegant solution for better research presenting the obtained results.

- Please, include correlation analyses for these results.

Reply: The advanced mathematical models proposed by the reviewer have been used in some papers (Zhang et al. 2014, Liu et al. 2010) to optimize the effects of processing parameters on polysaccharide extraction yield. In this respect, is it important to note that the aim of our work was not to investigate on the best conditions of extraction to increase the yield of polysaccharides, but to study whether an enriched fraction of polysaccharides (EPF), extracted from the fruiting body of Pleurotus erygii, possessed antioxidant activity and biocompatibility towards an astrocyte cell line, in the hypothesis of using polysaccharides as functional food for the prevention or complementary treatment of various pathologies including neurological diseases. For this reason, we used an established method of extraction of polysaccharides and tested the antioxidant activity, the cell viability and the ability of polysaccharides to counteract ROS production on the astrocyte cell line. Having prepared the EPF using fixed parameters for the extraction of polysaccharides, we cannot do correlation analyses for these results.

Therefore, we thank the referee for his/her valuable suggestion to apply advanced mathematical models to reinforce the results of this work, but we will welcome this suggestion for future studies to verify whether different conditions, applied to our extraction method, can allow to select the best conditions to increase the yield of polysaccharides extracted from Pleurotus eryngii.

- In the Methods part, include the reference for the reducing power assay.

 Reply: we have added the reference for the reducing power assay [71].

- The whole text needs to be revised (technical issues – coma, point, etc.).

- English editing required.

Reply: We have revised the whole text as required.

Reviewer 2 Report

The paper titled “Antioxidant activity of polysaccharides from the edible mushroom Pleurotus eryngii is very interesting and well written. I recommend the publication after a check of English language

Author Response

Reply to the reviewer 2

Review Report Form

English language and style

( ) English very difficult to understand/incomprehensible
( ) Extensive editing of English language and style required
( ) Moderate English changes required
(x) English language and style are fine/minor spell check required
( ) I don't feel qualified to judge about the English language and style

Yes

Can be

 improved

Must be improved

Not

applicable

Does the introduction provide sufficient background and include

all relevant references?

( )

(x)

( )

( )

Are all the cited references relevant to the research?

( )

(x)

( )

( )

Is the research design appropriate?

( )

(x)

( )

( )

Are the methods adequately described?

( )

(x)

( )

( )

Are the results clearly presented?

( )

(x)

( )

( )

Are the conclusions supported by the results?

( )

( )

( )

( )

Comments and Suggestions for Authors

The paper titled “Antioxidant activity of polysaccharides from the edible mushroom Pleurotus eryngii” is very interesting and well written. I recommend the publication after a check of English language.

English language and style are fine/minor spell check required

Reply: As requested, a revision of the English grammar has been made.

Reviewer 3 Report

In my opinion this manuscript is suitable for publication after the suggested improvements and after solving the dubious conclusion. 

My comments are:

1.     Revise the writing and the punctuation (e.g. line 66, 74);

2.     In scientific writing, avoid the use of first person (e.g.line 76, 81)

3.     Discuss glucan yield. Compare with previous studies using different analytical methodology. 

4.     Compare a- and b- glucan concentrations with studies using different assay.

5.     Can you explain why the chemical composition of the enriched polysaccharides fraction (EFP) is not 100% (Table 3)? What other components could be present? Compare with the literature.

6.     Antioxidant activity discussion is poor. Only presents data and states other works.

7.     The conclusion is inconsistent with the results. High content in phenolic compounds line 435-436? Or phenols not found (Table 3)?

Line 45, check reference 14 

Line 66, check spacing

Line 74, check grammar

Line 77, check reference 

Line 103, enriched or crude polysaccharide fraction (EPF)? Revise.

Line 148, Table 3, what is the CFP acronym?

Line 156, Table 4, indicate number of replicates

Line 177, Figure 2, superoxide radical, check chemical formula

Line 180-185, this description should go to discussion text. Fig. legend should be lean.

Line 192, RNS acronym?

Line 213-15, transfer; Fig. legend should be lean.

Line 229, CNS acronym?

Line 238, withdraw reference 44 as it is not correlated to antioxidant effect

Line 250-262, transfer; Fig. legend should be lean.

Line 273, “Calbiochem”, indicate where from ?

Line 306, correct “B-glucan”

Line 332, “(see par. 2.4)” means ?

Line 558, capital letters missing in reagent name

Author Response

Reply to the reviewer 3

Review Report Form

English language and style

( ) English very difficult to understand/incomprehensible
( ) Extensive editing of English language and style required
( ) Moderate English changes required
(x) English language and style are fine/minor spell check required
( ) I don't feel qualified to judge about the English language and style

Yes

Can be

improved

Must be improved

Not

applicable

Does the introduction provide sufficient background and include

 all relevant references?

( )

(x)

( )

( )

Are all the cited references relevant to the research?

( )

(x)

( )

( )

Is the research design appropriate?

(x)

( )

( )

( )

Are the methods adequately described?

(x)

( )

( )

( )

Are the results clearly presented?

( )

(x)

( )

( )

Are the conclusions supported by the results?

( )

( )

(x)

( )

Comments and Suggestions for Authors

In my opinion this manuscript is suitable for publication after the suggested improvements and after

solving the dubious conclusion. 

My comments are:

  1. Revise the writing and the punctuation (e.g. line 66, 74);

Reply: done 

  1. In scientific writing, avoid the use of first person (e.g.line 76, 81);

Reply: done 

  1. Discuss glucan yield. Compare with previous studies using different analytical methodology. 

Reply: We thank the reviewer for his/her suggestions and we have now added more comments in the paragraph 2.3, comparing our results with those reported in different studies. In particular, the yield of polysaccharides obtained in our study has been discussed in relation to the results obtained in other studies performed using different analytical techniques.

Line: 171-188 the following sentences have been added: “As shown in different studies, the yield of polysaccharides extracted from P. eryngi may vary in response to the method used for their extraction. As reported by Barbosa et al [46], in addition to the hot aqueous extraction, which represent the most widely used technique for polysaccharide extraction [47], there are other polysaccharide extraction techniques such as microwave assisted extraction, pulse extraction, ultrasound assisted extraction, alkaline extraction, enzyme-assisted extraction, subcritical water extraction and the use of deep eutectic solvents, each of them having advantages and disadvantages. The extraction yield obtained in this study is slightly lower than that reported by Xu et al [48] whereas it is much higher when compared with the 7.31% and 12.18% obtained by Li and Shah [49] and by He et al [50], respectively. In the first two cases, the polysaccharides extraction was performed with hot water (100°C x 120 min and 95°C x 180 min) whereas in the latter by hot alkaline extraction (90°C x 300 min). Based on these data, it would appear that the procedure used in this study, based in sequence by hot water and alkaline extractions, allows to obtain a higher yield of polysaccharides. However, it is worth mentioning that Zhang et al [22], using a response surface methodology to study the effects of ultrasonic power, ultrasonic time and ratio of material to liquid on the extraction yields of P. eryngii polysaccharides, reported a yield of P. eryngii polysaccharides reaching 34%”.

  1. Compare a- and b- glucan concentrations with studies using different assay.

According to the suggestion of the referee we have added a comment in paragraph 2.2 of the revised version of the manuscript in which we compared our results on the concentration of a- and b-glucans with those found in other studies using different assays.

Lines: 108-125 the following sentences have been added: “However, the glucan content may vary according to the topological specificity in the fruit bodies, the individual strains and the assay used for its detection. Different methods have been described to measure the content of glucans extracted from mushrooms. Mc Cleary and Draga [42] compared different procedures and developed a reliable, quantitative method for the measurement of β-glucans in mushroom and mycelial products. This method is based on controlled acid hydrolysis of total glucans, measurment of α-glucans using an enzymatic procedure and detection of β-glucans by difference. In comparison with other enzymatic procedures for the direct measurement of β-glucans, the method described by Mc Cleary and Draga [42], which reseambles the method used in the present work, allowed to detect in their samples from P. eryngii amounts of total and β-glucans higher than those detected in our samples, whereas the amount of α-glucans was smaller. By contrast, Manzi et al. [16, 25] described another procedure based on the direct hydrolysis by lichenase and β-glucanase and a content of β-glucan in the fruit bodies of P. eryngii about 1–2 order lower than that measured in our samples. These authors attributed the low glucan content to the presence of inert, insoluble material in the fibre residue, which prevents the diffusion of enzymes during the β-glucan measurement”.

  1. Can you explain why the chemical composition of the enriched polysaccharides fraction (EFP) is not 100% (Table 3)? What other components could be present? Compare with the literature.

Reply: A possible reason why the chemical composition of the enriched polysaccharides fraction (EFP) did not reach 100%, may be the presence in the fraction of small amounts of other minor compounds such as sulphate compounds. In this respect, the presence of small amounts (1.85 ± 0.28%) of sulphate compounds was reported by Zhao et al., 2020 (ref. 30 in the text) in the analysis of the chemical characteristics of polysaccharide fraction obtained from. P. eryngii.

  1. Antioxidant activity discussion is poor. Only presents data and states other works.

Reply: the discussion on the antioxidant activity has been expanded and our data have been compared with those obtained by other authors. (Paragraph 2.4, lines: 216-224 and 231-248).

  1. The conclusion is inconsistent with the results. High content in phenolic compounds line 435-436? Or phenols not found (Table 3)?

Reply:  We have completely revised the conclusion.

Line 45, check reference 14 

Reply: done

Line 66, check spacing

Reply: done

Line 74, check grammar

Reply: done

Line 77, check reference 

Reply: done

Line 103, enriched or crude polysaccharide fraction (EPF)? Revise.

Reply: done

Line 148, Table 3, what is the CFP acronym?

Reply: The correct acronym is EPF instead of CPF

Line 156, Table 4, indicate number of replicates

Reply: done

Line 177, Figure 2, superoxide radical, check chemical formula

Reply: done

Line 180-185, this description should go to discussion text. Fig. legend should be lean.

Reply: done

Line 192, RNS acronym?

Reply: The meaning of the acronym RNS, corresponding to reactive nitrogen species, has been added to the text

Line 213-15, transfer; Fig. legend should be lean.

Reply: done

Line 229, CNS acronym?

Reply: The meaning of the acronym CNS, corresponding to central nervous system, has been added to the text.

Line 238, withdraw reference 44 as it is not correlated to antioxidant effect

Reply: this reference has been replaced with the ref 62 in the revised version of the manuscript

Line 250-262, transfer; Fig. legend should be lean.

Reply: done

Line 273, “Calbiochem”, indicate where from ?

Reply: done

Line 306, correct “B-glucan”

Reply: done

Line 332, “(see par. 2.4)” means ?

Reply: corrected as “(see par. 3.4)”

Line 558, capital letters missing in reagent name

Reply: done

Round 2

Reviewer 1 Report

Thank you for the answers and comments. Manuscript can be accepted in the present form.